# Ectrodactyly with Polydactyly in a Dog—Case Description and Description of Surgical Therapy with Resection and Fusion Podoplasty

**DOI:** 10.3390/ani14111647

**Published:** 2024-05-31

**Authors:** Paul Wehrenpfennig, Philipp Schmierer

**Affiliations:** 1Veterinarian Clinic Posthausen, Posthausen, 28870 Ottersberg, Germany; 2DECVS, Veterinarian Clinic Posthausen, Posthausen, 28870 Ottersberg, Germany

**Keywords:** ectrodactyly, polydactyly, metacarpal bones, fusion podoplasty, surgery, dog

## Abstract

**Simple Summary:**

We report a rare congenital deformity of the phalanges of the forelimbs in a 3.5-month-old mixed-breed dog, causing lameness. A unilateral ectrodactyly with hypoplastic polydactyly of the third digit was surgically treated by partial amputation of excess tissue and stabilisation of the proximal metacarpal bones with a cortical screw in compression and a K wire. Three months after the initial surgery, metal fatigue failure necessitated the removal of the implant; however, the dog achieved near physiological limb function three months thereafter. This single case report adds to the knowledge of congenital limb deformities and their treatment options.

**Abstract:**

Objective: To describe a rare congenital deformity of the phalanges and the surgical details and outcome in a dog with ectrodactyly combined with polydactyly. Study design: Single case report. Animal: A 3.5-month-old male intact mixed breed dog with forelimb lameness and paw malformations. Methods: Surgery was performed on a dog with a congenital limb deformity consisting of resection of the extra bone and soft tissue structure to prevent further subluxation of the remaining metacarpals. Stabilisation consisted of a cortical screw in compression and a K wire across the proximal metacarpals. Results: Postoperative radiographs showed adequate implant positioning and good reduction of the proximal metacarpal row. At six weeks, the dog showed improvement in limb function and weight bearing. Major complications occurred at twelve weeks, and revision surgery with implant removal was required. At six months, the dog showed near normal range of motion and no lameness. Conclusions and Clinical Relevance: The decision to perform surgery on a dog with limb deformity resulted in an almost physiological gait, and the dog showed no abnormalities in daily life. This report adds to the literature on congenital limb deformities by describing the combination of ectrodactyly and polydactylism in a canine species, including the surgical approach and outcome. However, the optimal management of this heterogeneous condition is currently unclear.

## 1. Introduction

Dysostoses are described as rare congenital deformities of the appendicular skeleton [1,2]. The exact cause of these deformities is unknown, but proposed mechanisms include a failure of the mesenchymal bone model to convert precursor cells into cartilage or a failure to convert originally cartilaginous bone into bone [2]. Ectrodactyly (split hand, oligodactyly or hypodactyly), a type of dysostosis, is characterised by a paraxial longitudinal deficiency of one or more of the individual elements of the distal end of the limb [2,3,4]. It is the most common congenital malformation affecting the distal limb. The first and second metacarpals are most commonly affected [3]. The cleft is characterised by hypoplasia or the absence of one or more digital rays in the distal part of the forelimb [3]. In addition, more proximal parts of the limb, such as the carpal bones or the ulna, may be affected [2,5]. In dogs, ectrodactyly has been described exclusively in the thoracic limb and is usually unilateral [6,7], with only two reports describing bilateral occurrence [4,8].

Clinical signs range from mild deformity without lameness to more severe deformity with non-weight-bearing lameness [5]. Worsening with age and increasing secondary changes may occur. Treatment of ectrodactyly depends on the severity of bone and soft tissue abnormalities, clinical signs and lameness, but there is no general consensus in the literature [5,7]. Suggested treatment options range from conservative management to surgical correction. Information on surgical treatment is sparse in the current literature. Surgical treatment can vary from metacarpal fusion, partial or complete carpal arthrodesis or ulna osteotomy/ostectomy depending on the extent of the bone abnormalities [7,8].

Polydactyly is characterised by the presence of one or more extra digits and occurs as preaxial polydactyly on the medial side of the limb and post-axial polydactyly on the lateral side of the limb [9]. A preaxial malformation of the hind limb is commonly referred to as “dewclaw” and is described as a type of dominant genetic alteration [10].

To the authors’ knowledge, there is no publication in the veterinary literature with a combination of ectrodactyly and polydactyly, known as an ectrodactyly-polydactyly syndrome in humans. The purpose of this case report is to describe the surgical treatment, improvement in lameness and functional gait of a dog with ectrodactyly associated with polydactyly.

## 2. Case Details

A 3.5-month-old intact male mixed-breed dog weighing 24 kg was presented to the orthopaedic service of the small animal hospital in Posthausen because of lameness in the forelimbs and a deformed paw, as reported by the owners. On presentation, the orthopaedic examination revealed grade II/IV lameness of the right forelimb without any pain on palpation. The dog had hyperextension of the distal limb and a severe splayfoot. Palpation of the right forelimb revealed soft tissue abnormalities with absent interphalangeal skin and abnormal bony structures between the second and fourth metacarpals compatible with a split claw (Figure 1). Additional bony structures were palpable between the second and fourth digits, representing only one claw (distal phalanx) and lacking bone structure of the proximal and middle portions of both metacarpals (Figure 1). On examination and palpation of the abnormal claw, additional spreading of it was recognised, and a subluxation of the proximal aspect of the fifth metacarpal was also noted. Although initial ROM was not measured, the affected limb showed increased carpal ROM in extension compared to the unaffected side. The orthopaedic findings led to a tentative diagnosis of ectrodactyly associated with polydactyly of the third metacarpal of the right forelimb.

Dorso-palmar and medio-lateral radiographs of the right manus were obtained. The third digit showed a double conformation with only one claw. The third metacarpal showed an absent diaphysis with a proximal metaphysis, and two distal parts of the metaphysis and doubled phalanx were present. In addition, a severe valgus deformity of the second phalanx of the fourth digit with a varus deformity of the distal phalanx was observed in the presented case (Figure 2A).

There were no abnormalities of the carpus, antebrachium or elbow joint. The paw abnormalities seen in Figure 1, Figure 2A and Figure 3A supported the diagnosis of ectrodactyly and polydactyly of the third metacarpals. A physical examination of the dog revealed no other abnormalities or additional congenital anomalies.

Based on the clinical and radiographic findings, conservative and surgical options were discussed with the owners. As the patient did not show pain upon palpation of the pawn at the time of presentation, the mechanical irritation during weight bearing of the altered claw was thought to be the most likely cause of the lameness. In addition, the severe subluxation of the proximal part of the fifth metacarpal bone when spreading the splay foot was assessed as a potential reason for further deterioration. Lastly, the severity of the lameness and the abnormal position of the paw when standing led to the decision to perform surgical correction. The aim of the surgery was to remove the additional bone and soft tissue structure of the cleft and to fuse the proximal part of the remaining metacarpals (Figure 2A–F).

On the day of surgery, the dog was premedicated with medetomidine (10 μg/kg, Domitor 1 mg/mL, Vétoquinol) and methadone (0.2 mg/kg, Comfortan 10 mg/mL, Dechra). General anaesthesia was induced with propofol (2 mg/kg, Narcofol, CP-Pharma) and maintained with isoflurane in oxygen in a closed system after endotracheal intubation. Cefazolin (22 mg/kg, Cefazolin, Sandoz) was administered perioperatively.

After aseptic preparation, a dorsal approach was made to the third metacarpal (Figure 2B) with the patient in dorsal recumbency. The limb was extended caudally to facilitate the dorsal approach. Starting with a skin incision, the excess digits were isolated with careful blunt and sharp dissection to preserve the vascular structures as much as possible (Figure 2C). After successful isolation, the bony parts were resected (Figure 2D). After complete resection, the carpometacarpal joint space was carefully identified using a 25 G hypodermic needle. Two-point reduction forceps were placed over the proximal metacarpals to reduce the space between the proximal parts of the remaining metacarpals (Figure 2E). A cortical screw (2.0 mm) and a 1.8 mm K wire were then placed across the proximal metacarpals (Figure 3B). The surgical site was irrigated with sterile saline and closed routinely (Figure 2F). Post-operative orthogonal radiographs showed adequate implant positioning and good reduction of the proximal metacarpal row (Figure 3B).

After an uneventful recovery, the dog was discharged from the hospital on robenacoxib (2 mg/kg, Onsior, Novartis) for one week post-operatively. The operated forelimb was dressed with a splint bandage for three weeks, followed by a modified Robert Jones dressing for a further ten days. The owner was instructed on aftercare, including confinement and short leash walks 3 to 4 times a day for four weeks. Wound checks and initial dressing changes were performed by the referring vet with no concerns. Twelve days post-operatively, the owner reported adequate weight bearing and no obvious complications. Owners were educated about the importance of post-operative rehabilitation with a certified rehabilitation specialist. Suggestions to the owner included cold compression therapy in the post-surgical period (first four days), lymphatic drainage per massage and gentle ROM exercise in the beginning. The owners were instructed that the dog should receive professional physiotherapy 1–2 times per week for the first 4 weeks and two times per month for the following 4 weeks, depending on the healing process.

At the six-week follow-up, the patient was presented to our hospital for orthopaedic re-evaluation, including evaluation in stance and ambulation, palpation of the limb, goniometry, and radiographic examination. The dog was in good physical condition. Orthopaedic examination revealed lameness grade I/IV of the right forelimb and no pain was elicited on palpation. The range of motion (ROM) of the operated limb showed an increased angle of extension with a possible carpal extension of 215° and a slightly decreased carpal flexion angle of 42° compared to the described normal ROM of the canine carpal joint (196°/32°) [11]. The sound limb showed an almost normal ROM (195°/30°). Radiographs showed unchanged implant position and incipient fusion of the proximal aspect of the metacarpals (Figure 3C).

At 12 weeks, the dog presented with a sudden onset of severe lameness and was referred to our hospital. Orthopaedic examination revealed grade III/IV lameness and pain on palpation over the proximal metacarpus. Radiographs showed a fracture of the trans-metacarpal screw and migration of the K wire (Figure 3D). Cyclic overload at the transition of the fourth to the fifth metacarpal is most likely the reason for screw breakage with consecutive loosening of the smooth pin after increasing motion between the two metacarpals. Implant removal was discussed with the owners, and the dog received robenacoxib (2 mg/kg, Onsior, Novartis) until revision surgery due to severe discomfort.

After implant removal, post-operative radiographs showed no change in the position of the metacarpals compared to the initial post-operative radiographs (Figure 3E). Four weeks after the removal of the implants, an orthopaedic examination revealed no lameness on ambulation and no pain response upon palpation, but with lameness grade I/IV on extensive exercise reported by the owners. The range of motion (ROM) of the right carpus showed 208° of extension and 39° of flexion.

At the six-month follow-up after the initial surgery, there was no longer any lameness at a walk or trot (Figure 3F). However, the owners still reported grade I/IV lameness after extensive exercise. On palpation, the dog showed no signs of pain, and the carpal joint had an almost normal ROM of 204°/37°. The final radiograph was obtained during spaying 14 months after the initial surgery (Figure 3G), and no lameness was noted by the owners even after extensive exercise. Regarding the abnormal posture of the fourth toe, the dog showed no abnormalities, such as pain upon palpation or splintering at any of the examinations.

## 3. Discussion

This case report describes the clinical and radiological features of appendicular skeletal dysostosis consisting of a combination of ectrodactyly and polydactyly. To the authors’ knowledge, the combination of these two anomalies and their surgical treatment has not been described before. The nomenclature of abnormal limb deformities in veterinary medicine is not well defined. Polydactyly is defined as the presence of one or more additional digits. In the presented case, an additional hypoplastic metacarpal bone with the respective deformed and incomplete phalangeal bones was observed.

Ectrodactyly is well described in humans and other species [12,13,14]. In most cases of ectrodactyly, this type of dysostosis is combined with other anomalies such as ectodermal dysplasia, cleft palate and thumb-with-three phalanges [15]. In embryological development, there are three parallel limb rays—the medial ray forms the radius, the associated carpal bones and the first digit; the middle ray is responsible for the carpal and metacarpal bones and the second digit; the lateral ray forms the ulna, the associated carpal and metacarpal bones and the phalanges of digits III, IV and V [5]. In the described case, with the split cleft of the third digit and the absence of the metacarpals, an abnormal development of the lateral ray seems most likely. There is currently no specific information in the literature regarding the pathogenesis of ectrodactyly in dogs [13,16]. In humans and the dactylaplasia mouse, the pathogenesis of ectrodactyly is based on the hypothesis of a missing central ray. The combination of ectrodactyly and polydactyly, known as an ectrodactyly-polydactyly syndrome in humans [14,17], is a rare genetic limb malformation characterised by hypoplasia or absence of the central digital rays of the hands and/or feet. The presence of one or more unilateral or bilateral supernumerary digits on the post-axial rays ranges from hypoplastic digits without bony structures to complete duplication of a digit. The presence of polydactyly might even be the cause of the ectrodactyly [17].

There is no specific treatment recommendation for dogs with ectrodactyly. In human medicine, the main goals of surgery are to improve the child’s grasping and pinching abilities and to provide aesthetic reconstruction [12]. Depending on the deformity, surgical options in humans include skin separation, synostosis, phalangeal transfer and bone lengthening [18]. Surgical corrections described in the human literature are unlikely in veterinary cases due to the functional differences of the limb [16]. In veterinary medicine, treatment depends on the type and severity of the congenital limb deformity. In dogs, conservative or surgical management is described [19]. The primary goals are to prevent disease progression and to improve quality of life [8,20]. In the case of surgical management, the primary goal is to achieve metacarpal synostosis and restore limb function [5,8]. In the present case, the ectrodactyly was combined with a polydactyly, aggravating the distance between the adjacent metacarpals and even leading to proximal subluxation of the fifth metacarpal. Stabilisation of the remaining metacarpals with a cortical screw in combination with a K wire after resection of the abnormal cleft and redundant bone and compression with two-point reduction resulted in an excellent clinical outcome in the reported case. We decided not to remove the rudimentary part of the metacarpal bone of the third toe in order to limit soft tissue trauma. To accelerate the process of synostosis in this case, the use of a corticocancellous bone graft, as described in a previous study [20,21], would have been beneficial. However, due to the extensive resection of the abundant tissue and bone and fixation under compression, rapid healing was expected. The combination of pin and screw is an adequate surgical method to stabilise the metacarpals in such cases [5]. In the presented case, a larger cortical screw (2.4 mm) may have increased stability. However, in the presented case, with a diameter of less than 5 mm in the proximal metacarpus and the soft bone in this very young animal, the decision was made to use a 2.0 mm screw. It is disputable if using a 2.4 mm screw offering a higher area moment of inertia could have avoided the occurring complications. It is unknown if similar results would have been achieved without surgical treatment. However, due to the good function with reduced and non-progressive subluxation of the fifth metacarpal bone in the follow-up examinations, we believe that surgical therapy was beneficial in the presented case.

Complications following surgical correction of abnormal limb deformities can be various and reported complications of surgical treatment of ectrodactyly range from residual lameness to implant failure and removal [8,21]. In the reported case, a major complication of pin migration and transmetacarpal screw fracture at the junction of the fourth and fifth metacarpals was noted at the 12-week follow-up. As there was radiographic evidence of consolidation and no instability on palpation at the time of implant removal, placement of new implants was not performed. Twelve months after implant removal, the dog showed no lameness, even after extensive weight-bearing and physiological function of the limb. The authors consider fusion podoplasty to be an essential step in soft tissue reconstruction and functional restoration of the metacarpal pads, leading to full function. The goal to reduce the deformity of the fourth digit with remodelling after the fusion podoplasty was not successful. Fortunately, the dog did not show any problems in any of the follow up examinations. It is, however, unclear if osteoarthritis occurring at a later time point might affect clinical outcomes.

This single case report adds to the current literature with a combination of appendicular skeletal dysostosis consisting of a combination of ectrodactyly and polydactyly. The limitations of this report are those inherent in the nature of a prospective analysis of a single case report. In addition to the nature of a single case report, other limitations are that the outcome assessments may have been susceptible to inaccuracies due to the subjective perspective of the clinicians involved. The use of more standardised objective assessment methods, such as repeated measurements of gait analysis, forelimb muscle volume, pain score assessment and a validated owner questionnaire, would have provided a more consistent objective assessment. Another limitation is the lack of goniometry at the first examination, which would have provided a good basis for comparison. Unfortunately, at the time of the evaluation of the case there was no gate analysis available in our hospital. Stance- or gate analysis would have further helped to obtain objective outcome measures. An additional limitation was the short duration of follow-up to observe potential further complications such as recurrent lameness, changes in range of motion and possible problems with osteoarthritis.

## 4. Conclusions

This report adds to the literature on congenital limb deformities by describing the combination of ectrodactyly and polydactylism in one canine species, including the surgical procedure and outcome. The described surgical therapy led to normal function and no further subluxation of the metacarpal bones. However, due to the limited number, the optimal management in such cases of this heterogeneous disease is not yet clear.

## Figures and Tables

**Figure 1 animals-14-01647-f001:**
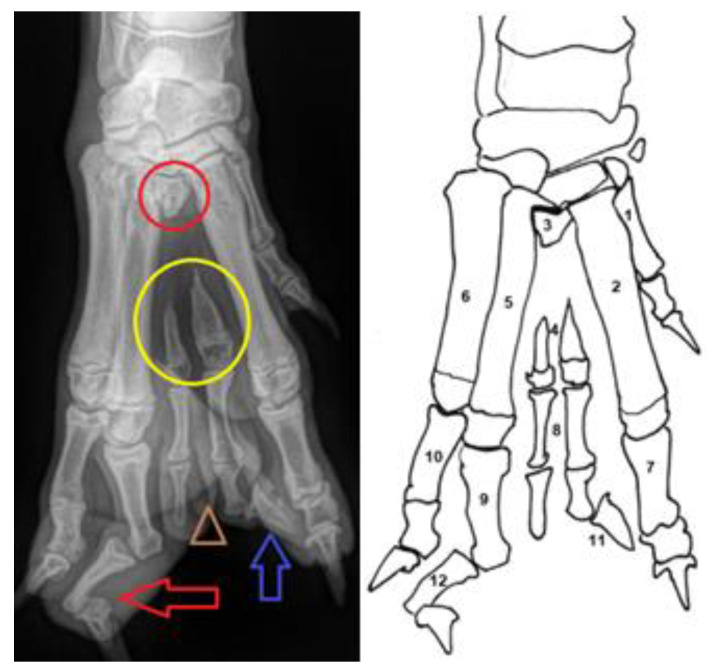
Pre-operative Radiographs (**left**) and line drawing (**right**) of the abnormal paw. (1) first metacarpal bone; (2) second metacarpal bone; (3) rudimental proximal part of the third metacarpal bone (the red circle); (4) hypoplasia of the third metacarpal bone (yellow circle); (5) fourth metacarpal bone; (6) fifth metacarpal bone; (7) proximal phalanx of the second digit; (8) double confirmation of the proximal and medial phalanx of the third digit (marked with the brown triangle); (9) proximal phalanx of the fourth digit; (10) proximal phalanx of the fifth digit; (11) note that only a single claw was developed at the third digit (blue arrow); (12) medial phalanx of the fourth digit with severe valgus deformity and distal phalanx of the fourth digit with varus deformity (red arrow).

**Figure 2 animals-14-01647-f002:**
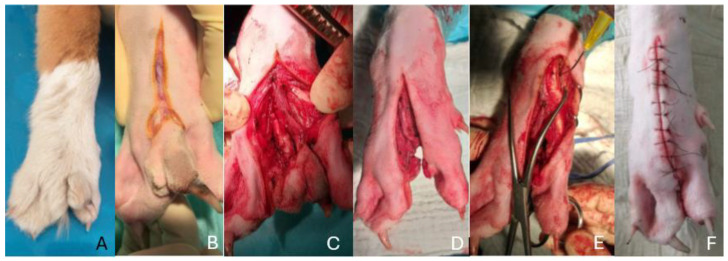
(**A**–**F**): Images sequencing the surgical procedure—(**A**) pre-operative image showing the dog’s paw. Note the gap between the second and fourth metacarpal bones compatible with the split claw. (**B**) A reverse Y skin incision was made, (**C**) the excess digits were carefully isolated using blunt and sharp dissection, (**D**) a post-operative image after complete excision was achieved, with (**E**) two-point reduction forceps used to realign the proximal metacarpals. (**F**) Post-operative image showing the surgical result after wound closure.

**Figure 3 animals-14-01647-f003:**
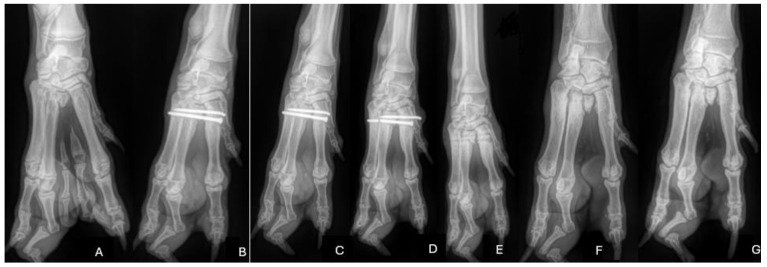
(**A**–**G**): Pre-operative (**A**) and post-operative (**B**–**G**) radiographs of the described case. (**A**) Pre-operative radiographs showing the metacarpal cleft and bony abnormalities. Note the third digit showing a double conformation with only one claw. (**B**) Post-operative radiographs—resection of the cleft was performed. A screw and K wire pin was used to stabilise the proximal metacarpals. (**C**) Follow-up radiographs 6 weeks after surgery showing no change in the position of the implant. (**D**) Follow-up radiograph 12 weeks after surgery showing a broken screw and migration of the pin. (**E**) Post-operative radiograph after implant removal. (**F**) Radiographs 3 months after removal and (**G**) 14 months after the initial surgery—there is no evidence of subluxation in the proximal region of the metacarpals.

## Data Availability

Data is contained within the article and dataset available on request from the authors.

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
