# Peer review of "Ectrodactyly with Polydactyly in a Dog—Case Description and Description of Surgical Therapy with Resection and Fusion Podoplasty"

_animals, 2024, doi:10.3390/ani14111647_

Round 1

Reviewer 1 Report

Comments and Suggestions for Authors

The work is very interesting, but I recommend making some corrections to improve its quality.

Line 3: Replace “Fusionpodoplasty” for “fusion podoplasty”.

Line 8: Replace “dog” for “mixed breed dog”.

Lines 14-15: I would eliminate "major complication". Minor complications refer to issues that arise in the postoperative period but do not necessitate a revision of the surgical procedure. I think it would be better to indicate that a minor complication occurred, specifically that the migration of the needle and the breakage of the screw observed at 20 weeks led to their removal.

Line 21: Replace “Surgery” for “surgery”.

Lines 22-23: Although the technique is explained in detail later, I would indicate at this point that the screw was placed in compression.

Line 33: Add the keyword "surgery", and if only 4 words are allowed remove “fusion”. Replace “fusion” for “fusion podoplasty”.

Line 45: Replace “fore limb” for “forelimb”.

Lines 65-66: In humans, the joint presentation of these congenital deformities is part of a syndrome known as ectrodactyly-polydactyly syndrome. Have the authors not considered using this terminology?

Lines 105-106: In the following sentence the authors indicate how the animal is positioned “After aseptic preparation, a dorsal approach to the third metacarpal bone was per-105 formed (Figure 1 B) with the patient in dorsal recumbency”. In this position, the limb should be placed in extension caudally to make a dorsal approach. The authors must explain how the limb was positioned to perform the surgical procedure (For example: “The limb was extended caudally to facilitate the dorsal approach")

Lines 112-113: The authors explain how the screw placement was carried out to compress the proximal portion of the metacarpals. It should be noted in this section that the remainder of the third metacarpal was left intact. It would be valuable to present the results and discussion regarding the decision to preserve this portion, and why the techniques described by Pisoni et al. (2014) were not considered. While I am not suggesting that the procedure used in this study was incorrect, given the favorable clinical evolution, I propose that this citation and its technique be discussed in the results and discussion section of the study.

Line 113: Replace “2,0 mm” for “2.0 mm”.

Line 113: Replace “1.8mm” for “1.8 mm”.

Lines 118-120: I believe that when the authors refer to using a reinforced Robert Jones bandage, they are describing a splint. Conversely, when they mention a soft padded bandage, they are likely referring to a modified Robert-Jones bandage.

Lines 198-202: The authors indicate that 2.7 mm screws could be used to proximally fix the metacarpal bones, but the size in this patient made it impossible. I believe that in a 24 kg dog, screws with a diameter larger than 2.0 mm could have been used, and even a 2.4 mm screw could have been suitable. Could the authors provide the width of the metacarpals in the mediolateral projection of the operated case and indicate the percentage of occupation of the screw used?

Lines 226-230: Particularly in the conclusions, I would emphasize the significance of describing the treatment of this pathology, while briefly acknowledging some of the limitations of the study.

I believe that the authors should emphasize that one of the most important objectives of the surgical procedure is to improve the quality of life of the patient’s undergoing surgery. In the conclusions, it must be made clear that achieving anatomical recovery to normality is impossible, and the primary goal is functional improvement in the affected limb, as demonstrated in the case they have described.

Lines 241-285: The references need considerable improvement. The format in which they have been introduced is entirely different from the rest of the text, and the nomenclature is not consistent across different citations. The last quote is missing the numbering.

Reviewer 2 Report

Comments and Suggestions for Authors

Comments on:  Ectrodactyly with polydactyly in a dog….

Overall, the paper is interesting because it describes a rare congenital limb deformity in a dog. Yet, I do not recommend publication in the present form; conclusions on treatment outcome and on the basic surgical concept should be thoroughly revised. The manuscript should be re-written, and re-submission should be encouraged; some sections like the abstract and summary need thorough revision. 

That the deformity is an ectrodactyly AND a polydactyly seems disputable; it seems that the case exhibits an ectrodactyly with hypoplasia of the 3rd metacarpalia which are doubled and visible on radiographs ? Or not ? On the image 1 A it is not clear whether polydactyly was actually visible (a rudiment of the claw of the 3rd digit seems present – your textual description says so – I suggest labelling the photographs with arrows for better understanding). Recommend description of the anomaly in detail since the case features one of the many possible variations of ectrodactyly and as such is a valuable contribution. (Mentioning “dewclaw” seems out of context as this is rather an atavism than a polydactyly). Perhaps it might be prudent eliminating polydactyly from the title but discussing it in the text. A better radiograph with proper labelling of the abnormal structures should be included (perhaps even a drawing like those in no 14 of the cited publications). There is a severe valgus deformation of the 2nd phalanx of digit 4 (with varus deformation of phalanx 3). It remains unclear to what degree this abnormality was causing pain and/or dysfunction, especially because this 4th digit in this case is the main weight bearing structure; this is neither described nor discussed in the paper. Also, as it is known that ectrodactyly in dogs may be accompanied by other distant malformations such as abnormal vertebrae, cleft palate etc., statements to that effect should be included (not only in the Introduction), confirming that this was investigated and was not present.

The entire paper focusses on surgery; this is unnecessary if not wrong in my view, because of many reasons:

Surgery in the present case might not have been the best solution; there is no evidence that surgical treatment was beneficial; the dog was operated as puppy and with or despite surgery was nearly free of lameness when outgrown (about 7-8 months of age). The surgical concept was doomed to fail; what the authors describe as “major complication” was an expected metal fatigue failure of a transverse screw 3 months down the road. This, because the attempted fusion of the metacarpalia 2, 4 and 5 could not and did never happen as there was never bony contact (besides perhaps neighboring intact cortical bone). Fusion could not occur; implants were prone to fail in this physiologically unstable environment; nevertheless, the attempt to reduce the gap was perhaps beneficial by excess tissue resection, fibrosis, and scar formation; all this should be considered in the Discussion; also, when screening the few cases of ectrodactyly published previously, similar cases of isolated clefts (without carpal, elbow or humeral joint abnormality) were not treated surgically. This does not mean that surgery was contraindicated, it only means that a conclusion favoring surgery cannot be drawn in this case, also, considering the method chosen and the resulting complications.

If, in a revision of the manuscript, surgery is reported, then suggest keeping it short; skip unnecessary length of anaesthesia description, delete pre-op disinfection, eliminate terms like “careful preparation” (it is obvious that a surgeon operates with care ….); delete the name of the reporting clinic in the text – address of authors as reporting institution should suffice. The Kirschner wire which was inserted should not be called “anti-rotational pin”; there is no rotation, there is perhaps distraction; the term comes from the technique of fixing humeral condylar fractures and should not be used here.

Suggest re-writing Discussion and Conclusions based on above comments.   

Revised accordingly, the paper will be a valuable addition to the literature based on the relative rareness of the deformity which, however, warrants better description (and not because of treatment with ambiguous outcome).  

Ref. citations 2, 14 are incomplete.

To exemplify shortening text, I started editing the Short Abstract (as suggestion) – see below:

Paul Wehrenpfennig, Veterinary Clinic Posthausen, 28870 Ottersberg - Posthausen, Germany

Philipp Schmierer DECVS, Veterinary Clinic Posthausen, 28870 Ottersberg - Posthausen, Germany, *   Correspondence: to: Paul Wehrenpfennig, paul.wehrenpfennig@gmail.com, tel 0049/15750101831

Simple Summary: We report a rare congenital limb deformity of the phalanges on the front limb in a 3,5 month-old dog, causing lameness. A unilateral ectrodactyly with hypoplastic polydactyly of the 3rd digit was treated surgically by partial amputation of excess tissues and stabilization of the proximal row of the metacarpal bones using a cortical screw and K-wire. Three months after surgery, metal fatigue failure necessitated implant removal; however, 3 months thereafter, the dog gained nearly physiological limb function.

Comments on the Quality of English Language

short summary and abstract need thorough English language and textual revisions; the remainder of the text necessitates only small corrections

Reviewer 3 Report

Comments and Suggestions for Authors

General comments:

This is a very interesting study with some issues that should be addressed before publication. It would be interesting if the authors had chosen stance analysis and gait analysis. Limitations should state this and the absence of a claudication and pain score for evaluation. 

As only one case, it has to be detailed. Authors should mention rehabilitation modalities, exercises and progressive therapeutic approach of this case. Even if it wasn´t performed it should be discussed. 

Photos should be added. 

Also, it would be important to discuss and explain what was the cause of the major complication regarding pin migration. 

Specific comments:

Line 36, line 41 and 42, line 53: I suggest adding some more bibliographic references to these sentences;

Line 63-66: The objectives should be re-written because the main aim is not only the surgical approach but its viability to improve lameness and functional gait;

Line 71-72: If the patient has a grade II/IV lameness, what about pain?  Is this lameness due to pain, inflammation, or only from a mechanical origin? 

Line 118: Why not perform cryotherapy? Or other modalities to improve healing? Why not kinesiotherapy exercises and gait stimulation to correct limb position on the floor to promote weight bearing? These should be discussed.

Line 121: This is confusing, please explain better.

Line 123: It would be interesting if the authors could add a figure.

Line 130: Goniometry was not performed in the first consultation? It would be interesting to compare the measures.

Line 136: What was done for pain management?

Round 2

Reviewer 2 Report

Comments and Suggestions for Authors

Thank you for your thorough revision; the result is most satisfying and I recommend publication in the present form without hesitation; with one exception: the title; please change again; the repetition of the word "description" is somewhat awkward (and, ...."in one dog" - is generally understood as "case report" and a case report describes....) perhaps you should simply say: "Ectrodactyly with polydactyly in a dog – successful treatment by resection and fusion podoplasty". The drawing illustrating the malformation is excellent and is a valuable addition to the paper. Good work, congratulation. GN.

Reviewer 3 Report

Comments and Suggestions for Authors

The authors address all comments, and the manuscript is fine for publication.